# A weighted network analysis framework for the hourglass effect—And its application in the *C. elegans* connectome

Ishaan Batta[1], Qihang Yao[2], Kaeser M. Sabrin[2], Constantine Dovrolis[2]*

**1** School of Electrical and Computer Engineering, Georgia Institute of Technology, Atlanta, Georgia, United States of America, **2** School of Computer Science, Georgia Institute of Technology, Atlanta, Georgia, United States of America

* constantine@gatech.edu

## Abstract

Understanding hierarchy and modularity in natural as well as technological networks is of utmost importance. A major aspect of such analysis involves identifying the nodes that are crucial to the overall processing structure of the network. More recently, the approach of hourglass analysis has been developed for the purpose of quantitatively analyzing whether only a few intermediate nodes mediate the information processing between a large number of inputs and outputs of a network. We develop a new framework for hourglass analysis that takes network weights into account while identifying the core nodes and the extent of hourglass effect in a given weighted network. We use this framework to study the structural connectome of the *C. elegans* and identify intermediate neurons that form the core of sensorimotor pathways in the organism. Our results show that the neurons forming the core of the connectome show significant differences across the male and hermaphrodite sexes, with most core nodes in the male concentrated in sex-organs while they are located in the head for the hermaphrodite. Our work demonstrates that taking weights into account for network analysis framework leads to emergence of different network patterns in terms of identification of core nodes and hourglass structure in the network, which otherwise would be missed by unweighted approaches.

## Introduction

Many networks in both technological and biological systems tend to follow a structure where a large number of inputs and outputs are connected by pathways through a small number of intermediate nodes in the network [1–3]. In such a hierarchical organization of a network, these intermediate nodes are typically of high importance as they form either the bottleneck or the core of the overall information flow in the network. This pattern of hierarchy has been observed in many networks in biological as well as techno-social systems [2, 4–6]. In the case of various biological networks, modularity and hierarchy are well known to help organisms function ceaselessly with better robustness despite external disturbances to the input and

**Data Availability Statement:** The data underlying the results presented in the study are available from https://wormwiring.org and https://www.wormatlas.org/neuronalwiring.html. The code

implementing the analysis along with results can be found in the repository: https://github.com/QihangYao/weighted-hourglass.

**Funding:** Research reported in this publication was supported by DARPA Grant: Lifelong Learning Machines (L2M) program of DARPA/MTO: Cooperative Agreement HR0011-18-2-0019 (https://www.darpa.mil/program/lifelong-learning-machines). The funders did not play any role in designing the study, collecting and analyzing the data, decision to publish, or preparing the manuscript.

**Competing interests:** The authors have declared that no competing interests exist.

output modules [7, 8]. In addition to this robustness for survival, modular and hierarchical networks are also helpful in evolvability [9], as certain peripheral (input/output) modules of less criticality in the information-flow of the network can be modified more effectively in response to prolonged external environment changes spanning across multiple generations of the organism [7, 10, 11].

In such scenarios, it is important to examine the set of nodes which are critical to the structure or functioning of the network. Many measures have been used in network science to analyze centrality structure in networks [12–15]. More recently frameworks characterizing the bow-tie structure have been developed to study networks that are organized in a hierarchically modular fashion and facilitate distributed information processing [16–18]. In the same line, [19] developed the hourglass analysis framework to study hierarchical dependency networks, especially for networks with a relatively higher number of inputs and outputs mediated through a much smaller set of intermediate modules. The hourglass effect has been observed in networks from various domains in biology including metabolism [16, 20, 21], neuronal structure for visual-cognitive tasks [22]. The hourglass framework identifies a set of core nodes (known as the $\tau$-core) in a source-target dependency network based on the path centrality metric [19], returning a small set of critical nodes through which most input to output pathways pass. Additionally, [19] also developed the H-score metric to quantify the extent to which a given network shows the hourglass effect.

Among the neuronal networks in biology, the connectome of the *Caenorhabditis elegans* (*C. elegans*) has been of particular interest. It is the only fully-mapped structural connectome among all organisms with a nervous system, enabling the study of structural properties of neuronal networks, many of which could be generic to systems as complex as the human brain [23]. Recent work has pointed out that the *C. elegans* nervous system can be viewed more as a distributed information processing system [24]. The aim to identify critical neurons in the connectome has thus focused on creating measures that take an integrated picture from the whole network while computing the critical neurons and analyzing the properties of the network as a whole [25]. Had performed the hourglass analysis on the *C. elegans* connectome by considering it as an information processing system with an environmental input received by the input nodes (sensory neurons), processed by an intermediate network of nodes (inter-neurons) to generate an appropriate output response through output nodes (motor neurons). The hourglass analysis framework was extended to handle feedback loops in the given network doing away with the assumption of the network representing the information processing system being a directed acyclic graph (DAG) [25]. However, all the studies analyzing the hourglass properties have been done using the unweighted hermaphrodite connectome of the *C. elegans* published by [23]. Recently [26], published a new connectome of the *C. elegans* with datasets for both male and hermaphrodite sexes along with the weights for each connection based on the number and size of neuronal synapses between a pair of neurons as seen in electron microscopy (EM) series.

In this paper, we firstly present a new framework to study the hourglass effect in weighted networks in the context of the *C. elegans* connectome. The importance of taking weights in a network into account is well known [27, 28]. For the first purpose of the weighted hourglass analysis, we develop a method named multi-edge transformation (MET) which involves performing hourglass analysis on a weight-based transformation of the original network. An important concept in identifying the "waist" or core nodes in the hourglass analysis is the path centrality metric [19, 25], which was earlier defined only for unweighted networks. The MET method presented in this paper redefines the path centrality of a node by taking into account the weights of edges along the source-target (sensory-motor) paths that traverse through the node. Thus MET assigns importance to nodes in the network based on not merely the number

of source-target paths passing through them but also the extent to which these paths contain highly weighted edges. Our results on the *C. elegans* connectome data show that the weighted version (MET) of the hourglass analysis identifies a more specific yet not fully overlapping set of core nodes (neurons) as compared to the unweighted version (UNW). We also show through randomization experiments that the hourglass effect exhibited by the weighted networks at hand is not due to chance, as the empirical network had a significantly higher H-score compared to the corresponding networks generated through the randomization procedure.

Secondly, we also present a comparison between the male and hermaphrodite connectomes in terms of their hourglass properties. By using both the unweighted and weighted versions of the hourglass analysis on the male and hermaphrodite connectomes, we show that the neurons that form the core of the network structure differ significantly between the male and hermaphrodite, both in location and function. Almost all of the neurons that form the core of hermaphrodite were located in the head and are involved in functions like information integration, behavioral response, locomotion, and commanding. However, a large portion of the core neurons of male connectomes are in the pre-anal ganglion and involved in various functions related to mating behavior. While [26] had also observed differences between the male and hermaphrodite connectomes in terms of dedicated sex neurons in the male for certain reproductive functions, our work shows that the sex-specific neurons in the male are actually at the core of network structure, and these interneurons are highly intervened in a lot of sensorimotor pathways.

Taking together, our work presents a way to analyze a given weighted network for a small set of core nodes through which most of the input-output pathways pass, making them essential for most sensorimotor activity in the specific case of neuronal connectomes. The application of this framework on the *C. elegans* connectome shows that there are significant differences in the way the male and hermaphrodite connectomes are organized, especially in terms of the set of neurons at the core of end-to-end neuronal pathways. This difference pattern between the sexes persists irrespective of whether the connectome is considered as a weighted or an unweighted network. Our work opens a new possibility of how modern network analysis frameworks can be used to understand subtle structural differences that may play a part in defining the differences in behavioral characteristics of organisms.

## Materials and methods

### Datasets/Connectomes

Datasets with neuronal networks of the *C. elegans* for both male and hermaphrodite sexes from [29] were used in this study, referred to as *maleCook* and *hermCook* in the paper. Directed networks were built from the dataset with all chemical synapse connections between pairs of neurons represented as directed edges, and the number of synapses in each connection was used as the weight of each edge. While the dataset also provides information about the size of the chemical synapses, it is yet unclear how to interpret the influences that the size of a chemical synapse has in the neuronal signal transmission, thus we interpreted only the number of synapses between neurons as edge weight, and built our following models based on this interpretation. Only non-pharyngeal neurons were used to create the networks with each neuron categorized into sensory (S), intermediate (I), and motor (M) neurons. The number of neurons in each category for both male and hermaphrodite connectomes is detailed in Table 1. Hereafter, the term connectome would refer to the sub-network consisting of only S, I, and M neurons, unless mentioned otherwise.

**Table 1. Number of nodes (neurons) and edges (connections) in the networks generated by the three connectomes.**

| connectome | $n$ | $m$ | $n_S$ | $n_I$ | $n_M$ |
|---|---|---|---|---|---|
| maleCook | 358 | 3905 (3388) | 137 | 113 | 108 |
| hermCook | 280 | 3565 (2999) | 83 | 81 | 116 |
| hermVarshney | 279 | 2194 (1864) | 88 (76) | 87 (84) | 119 |

Each edge represents the existence of chemical synpase connectivity between two neurons, while the edge-weights represent the number of chemical synapses between two neurons. The numbers in parenthesis are results after the removal of feedback edges from the network. $n$: number of nodes (neurons), $m$: number of edges (connections), $n_S$: number of sensory neurons, $n_I$: number of inter-neurons, $n_M$: number of motor neurons.

For comparison purposes, we also include a previously widely-used unweighted connectome from [23](termed as *hermVarshney*). The details of number of neurons and edges in the networks generated by these connectomes can be found in Table 1. *hermCook* consists of considerably larger number of edges than *hermVarshney*, and the weights of the edges unique to *hermCook* were found to be significantly weaker than the weights of all the edges in *hermVarshney* ($p$-value $<10^{-76}$), as shown in Fig 1.

## Preprocessing

There are no nodes that play a dual role (out of S, I, M) as per the provided data in [29]. Four neurons in the male connectome have been mentioned in the cell list but not present in the connectome, which include two male-specific neurons 'CA01' (inter) and 'CP00' and two sex-shared neurons 'BAGR' (sensory), 'ADAR' (inter). Neurons near the pharynx were previously found to form a self-contained, autonomously acting nervous system [26]. Therefore, we include only non-pharyngal neurons in the hourglass analysis. After these processes, one neuron in the male connectome, 'URYDL' (sensory) was found to be not connected to other remaining neurons. We exclude it in further analysis.

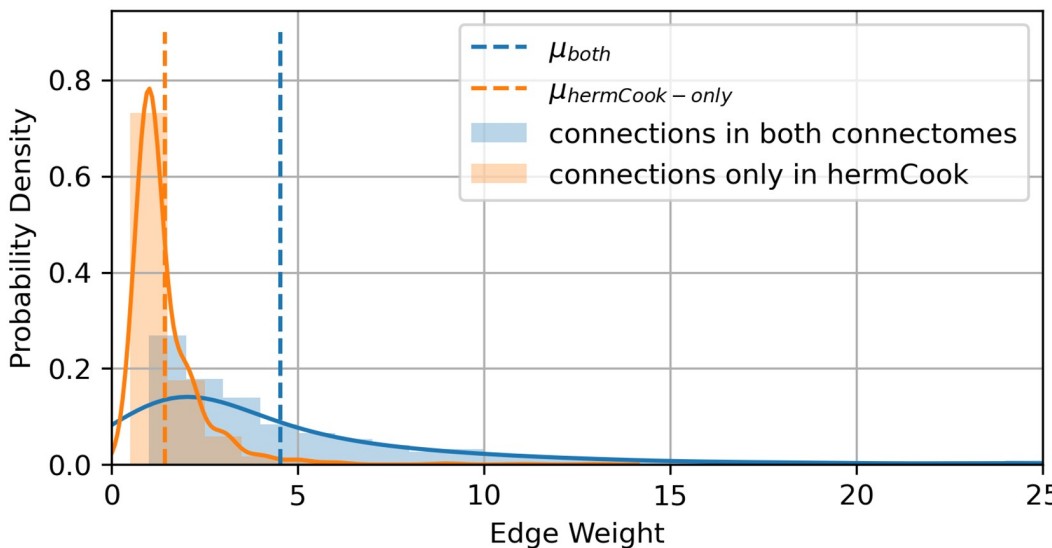

**Fig 1. Distribution of edge weights for the edges unique in *hermCook* and edges common to both networks.** Vertical dashed lines show $\mu_{int}$ (green) and $\mu_{uniq}$ (red), which are the average weights (count of synapses) of edges common to both *hermCook* and *hermVarshney*, and unique to *hermCook*, respectively. The additional edges in the *hermCook* connectome are significantly weaker ($p$-value $<10^{-76}$) than the edges that are common between both the connectomes.

**Table 2. Proportion of connections between sensory (S), inter (I) and motor (M) neurons for all connectomes.**

|  | S | I | M | S | I | M | S | I | M |
|---|---|---|---|---|---|---|---|---|---|
| S | 0.21 | 0.20 | 0.06 | 0.10 | 0.15 | 0.07 | 0.07 | 0.17 | 0.07 |
| I | 0.09 | 0.19 | 0.10 | 0.08 | 0.22 | 0.13 | 0.06 | 0.23 | 0.16 |
| M | 0.01 | 0.03 | 0.11 | 0.02 | 0.05 | 0.16 | 0.01 | 0.07 | 0.16 |
| (a) maleCook | | | | (b) hermCook | | | (c) hermVarshney | | |

Rows represent the category of the pre-synaptic neuron and columns represent the post-synaptic neuron of the chemical junction.

The details of number of edges for all connectomes after feedback edge removal can be found in Table 1. Table 2 shows the proportion of the 9 possible directed edge directions between *S*, *I* and *M* neurons. The feedback edges, which include $I − S$, $M − S$ and $M − I$ connections, were removed while doing the analysis. This was done to ensure efficient source-target path computation for the hourglass analysis [25].

The data provided in [29] also includes a separate section on sex-specific neurons. The male connectome has 84 sex-specific neurons leaving out muscles, categorized into sensory or inter neurons while the hermaphrodite connectome has only 8 sex-specific neurons (HSNL, HSNR, VC01, VC02, VC03, VC04, VC05, VC06), all of which are motor neurons.

## Hourglass analysis on unweighted networks

The hourglass analysis as described in [19] is a framework to analyze a network in terms of nodes that are responsible for most of the information flow between the source (or input) and target (or output) nodes of a network. If only a few nodes are a part of majority of the paths connecting the source and target nodes, then the network is said to exhibit hourglass property, and these nodes can be said to constitute the core of the network, similar to the waist of an hourglass. The following subsections detail out the mathematical formulation of this concept by defining the relevant metrics and methods to compute them.

**Generating source-target paths.**    If the *C. elegans* connectome is thought of as an information processing system to be analyzed in terms of the hourglass property, it can be done in terms of the number of source-target paths (S-T paths) i.e. paths connecting a source node to a target node in the network. Nodes through which a larger proportion of S-T paths pass can be considered as nodes of importance for the information processing system. Among all S-T paths present in a network, usually only a subset of the paths would be used in the information flow, which requires domain-specific knowledge to determine. In terms of *C. elegans* neuronal network, previous work [25] proposed a series of path constraints based on consideration of routing efficiency (top shortest paths) or diffusion efficiency (paths with limited lengths).

In this study, we used the set of S-T paths that are at most 4 hops in length. We restrict the set of paths in terms of maximum length rather than selecting the shortest ones because there is no biological evidence so far supporting the neuronal network's ability to route information through the shortest paths—which requires each neuron to have complete information about the connectome. Moreover, previous work on the *C. elegans* connectome evaluated a number of sensory-to-motor neuron path selection schemes—without noticing major differences [25]. We rely on this path selection scheme for brevity as it seems more plausible for neuronal communication. We use the notation $P_4$ to denote this set of paths. Hereafter, the term S-T paths will be used to refer to the set of paths from source to target nodes that belong to the $P_4$ paths.

**$\tau$-core and path centrality.**    There are numerous metrics for examining the extent to which a node plays a role in information transfer in the network. While metrics like degree

and eigenvector centrality are based on local properties, centrality metrics that are informed from the network in its entirety are also used for paradigms involving information flow. In the context of the *C. elegans* neuronal connectome, a relevant metric should account for the fact that the network function involves transfer of neuronal signals between various parts of the nervous system. We used the path centrality metric in this study for formulating the centrality of a node in the network. With the set of paths identified, the next step is to find the core of the network. This is equivalent to finding $\tau$-core, defined as the smallest set of nodes covering a fraction $\tau$ (the path-coverage threshold) of the entire set of paths. Unfortunately, finding $\tau$-core is an NP-Complete problem [19]. Nevertheless, by defining the concept of the path centrality we could build a greedy heuristic approach to compute the approximately optimal solution.

The path centrality $P(v)$ of a node $v$, also known as *stress* of a node [30], is defined as the number of S-T paths that pass through $v$. In the context of *C. elegans* connectome, the path centrality represents the number of sensory-motor neuronal pathways in which a particular neuron participates, thus capturing how important a neuron is to the overall neuronal transmissions in the network.

We could find an approximate solution for computing the $\tau$-core by iteratively adding nodes with the highest path centrality in the network into the core set, starting initially with an empty set. In each step $i$ after a node $v_i$ is picked, we compute all the paths in the $P_4$ that could be traversed by node $v_i$, and removed all the nodes and edges in this set of paths. Then, we recalculate the path centrality of each node before adding the next node. This is repeated until at least a fraction $\tau$ of S-T paths have been removed (covered). The entire process of obtaining the $\tau$-core using this iterative removal of nodes with highest path centrality is delineated more formally in Algorithm 1.

**Algorithm 1**. Computation of $\tau$-core using greedy heuristic

```
1: procedure ComputeCore(τ, P_ST)
2:     τ ← path-coverage threshold
3:     P_ST ← set of S-T paths
4:     n_p ← 0 (number of paths passing through core nodes)
5:     n_t ← |P_ST| (total number of S-T paths)
6:      C ← ∅ (set of core nodes initiated with empty set)
7:     while n_p < τ n_t do
8:        PC ← [P(v) ∀ v ∈ V, v ∉ C]      ▷ Path centralities of nodes
which are not in the core set
9:        bestNode ← argmax_v PC     ▷ Select node with the highest path
centrality
10:        C ← C ∪ {bestNode}      ▷ Add the node to set of core nodes
11:        P_best ← {p ∀p ∈ P_ST, bestNode ∈ p}      ▷ Create a set of paths
passing through bestNode
12:        n_p ← n_p + |P_best|
13:        P_ST ← P_ST\P_best ▷ Remove all paths passing through the newly
added core node
14:     end
15:      return C
```

**Hourglass score.** While the computation for the $\tau$-core of a network provides a set of nodes through which a large fraction ($\tau$) of S-T paths pass, to evaluate the hourglass property of a network we still need a reference with which the size of the core can be compared. This is because while the $\tau$-core can be computed for a very large $\tau$ close to 1, there is no gaurantee whether the size of this $\tau$-core is small enough to say that the network exhibits an hourglass effect when compared to a network that definitely does not. To resolve this, the H-score (or the Hourglass Score) has been used as a metric involving a comparison between the size of

$\tau$-core of the original network **G** with a derived network **G$_f$** known as the *flat dependency network*. **G$_f$** is constructed from **G** in such a manner that **G$_f$** has no hourglass effect while it captures the source-target relationships of **G** [19]. The procedure is as follows:

- **Nodes** The source and the target nodes are the same in **G$_f$** as they are in **G**. However, **G$_f$** does not have any intermediate nodes.

- **Edges** The edges for **G$_f$** are constructed by replacing each S-T path connecting each pair of source node *s* and target node *t* in **G** with an edge from *s* to *t* in **G$_f$**. Since this can lead to multiple edges between source node *s* and target node *t* as there can be multiple S-T paths from *s* to *t*, these edges can be replaced with a single edge of weight *w*, where *w* is the number of S-T paths between *s* and *t* in **G**.

It should be noted that the **G$_f$** is constructed in a manner such that the dependencies between source and target nodes are maintained along with the number of S-T paths between each pair of source-target nodes. However, it does not have any intermediate nodes (or set of nodes) through which most of the S-T paths would pass. In other words, **G$_f$** cannot exhibit hourglass property by design. Also, $\tau$-core of **G$_f$** cannot be larger than the $\tau$-core of **G**. This is because the $\tau$-core of **G$_f$** is made up of only source and target nodes while the $\tau$-core of **G** can have intermediate nodes replacing multiple source or target nodes in the core set. Therefore we can say that,

$$C(\tau) \leq C_f(\tau) \leq min\{|S|, |T|\} \tag{1}$$

where $C(\tau)$ and $C_f(\tau)$ are the sizes of the $\tau$-core of **G** and **G$_f$** respectively. *S* and *T* represent the set of source and target nodes respectively. The H-score is defined using this premise to measure the extent to which the network **G** shows the hourglass effect. The H-score $H(\tau)$ of a given network and threshold $\tau$ is defined as:

$$H(\tau) = 1 - \frac{C(\tau)}{C_f(\tau)} \tag{2}$$

It can be inferred that $0 \leq H(\tau) \leq 1$. The H-score is closer to one if the size of the $\tau$-core of the original network **G** is much smaller than that of **G$_f$**, thus implying that **G** shows stronger hourglass effect because of a smaller core set compared to **G$_f$**.

**Randomization experiment—Edge shuffling.** The H-score metric captures the extent to which the hourglass effect is present in a given network. However, we cannot rule out the possibility of the hourglass effect arising due to random artifacts, while not being a property of the topology of the particular network. To check this on unweighted networks, we performed a randomization procedure as in [25] for testing the statistical significance of the hourglass effect. This procedure shuffles the network edges while preserving total number of nodes and edges, in-degree of all nodes and the partial ordering of the nodes in the network [25].

## Hourglass analysis on weighted networks

The aforementioned hourglass analysis pipeline could be further extended by utilizing the edge weight information to provide a more accurate estimation of the hourglass property of a network. The core idea is that we could assign a weight to each path for estimating the relative importance of each path in the information flow in a network, followed by the computation of path centrality and $\tau$-core based on it. The following subsections detail the formulation of weighted hourglass analysis.

**Path weighting with multi-edge transformation.** For each path in the candidate paths identified through path selection ($P_4$ in this case), we assign a path weight based on the weight of edges in involved in the given path. Mathematically, we define the path weight as

$$w_p = f(w_{e_1}, \ldots, w_{e_n}), \quad e_1, \ldots e_n \in p \tag{3}$$

where $w_p$ is the weight of a path $p$, $e_1, \ldots e_n$ are the edges in $p$ and $w_{e_1}, \ldots, w_{e_n}$ denotes their corresponding weights. $f$ can be any function that accepts a randomly-large set of values and returns a single value. Different definitions of the edge weights may expect different definitions of $f$. For example, in networks where edge weight represents the flow through the edge, $f$ could be defined as taking the minimum among edges weights, so that the weight of the path represents the maximal possible flow along the path. If the weight of the edge represents the efficiency of communication between nodes, $f$ could be taking the inverse of the sum of the inverse of the inputs, so that the weight of path represents the communication efficiency between source and target.

In the case of *C. elegans* neuronal network, the weight of an edge represents the number of synapses in each connection. Therefore, we introduce the multi-edge transformation (MET) model, in which each edge in the network with weight $w$ is replaced of $w$ parallel edges of weight 1. As a result, each path $p$ identified in the original hourglass analysis will be counted $w_p$ times in the weighted hourglass analysis, with $w_p$ being the number of unique combinations of edges that could form the path. Mathematically, we have

$$w_p = \prod_{e \in p} w_e \tag{4}$$

Besides intuitive biological interpretation, the MET model has several advantages mathematically: 1) It can reduce to the original hourglass analysis when the weight of edges in the network is equal to 1; 2) The weight of any edges in a path is important for determining the weight of the path.

**Weighted path centrality and $\tau$-core.** If we interpret the path weight as the importance of a path, we could further modify the definition of core in hourglass analysis to be the minimum set of nodes that play a major role in the information flow in the neuronal network. In other words, we define $\tau$-core to be the smallest set of nodes that cover paths that explain more than $\tau$ ratio of total path weights in the network.

Again we can use the weighted path centrality to build a heuristic algorithm to compute the $\tau$-core. For a network with $P_{ST}$ as the set of S-T paths, we define the weighted path centrality of a node $v$ as

$$P(v) = \sum_{\substack{p \in P_{ST} \\ v \in p}} w_p \tag{5}$$

To visualize this concept, Fig 2 shows an example of a network and Table 3 values shows the path centrality computed for each of the nodes in this network, for both unweighted hourglass analysis and weighted hourglass analysis.

There may be nodes in the network which would not be a part of the core in an otherwise unweighted network with the same set of nodes and edges, but could still be responsible for most of the source to target pathways when the network is imagined in terms of the number of synaptic connections between neurons. Thus transforming the graph based on the weights i.e. number of synapses between neurons essentially captures this notion.

**Randomization experiment—Edge weight permutation.** To test the statistical significance of H-score against randomized networks [25], had done a randomization analysis in

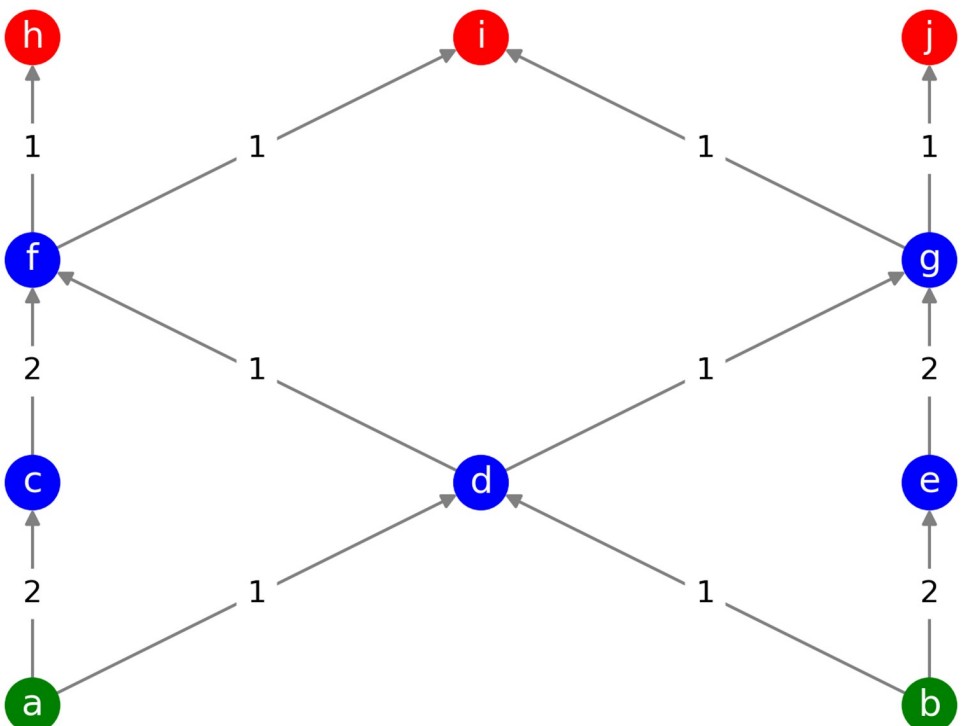

**Fig 2. A sample weighted network with 10 nodes.** Source, intermediate, and target nodes are represented in green, blue, and red colors respectively. It should be noted that node *d* has the highest path centrality as per the unweighted hourglass model since most of the paths pass through it. However, if the weights are considered and the multi-edge transformation (MET) model is used, then nodes *f* and *g* would be the nodes with the highest path centrality and forming the core.

their work which involved rearranging the network edges. For weighted networks, however, we could reduce the variance brought by randomization by using edge weight permutation. This can be done by shuffling the weight of the edges while keeping the structural topology of the network intact. This randomization allows us to more precisely evaluate the hourglass effects of the empirical network caused only by the weights instead of the network topology. We compared the empirical network with 500 such randomized networks generated by edge-weight permutation to check the statistical significance of the hourglass effect in the networks.

## Results

### Hourglass effect in weighted and unweighted networks

We apply the hourglass analysis according to the framework mentioned in subsection *Hourglass analysis on unweighted networks* using the routing scheme $P_4$ for generating source-target paths as described in subsection *Generating source-target paths*. Let *P* denote the set of paths

**Table 3. Path centrality of all nodes of the network in Fig 2.**

| method | a | b | c | d | e | f | g | h | i | j |
|---|---|---|---|---|---|---|---|---|---|---|
| unweighted | 6 | 6 | 2 | 8 | 2 | 6 | 6 | 3 | 6 | 3 |
| MET | 12 | 12 | 8 | 8 | 8 | 12 | 12 | 6 | 12 | 6 |

Path centrality was measured using the unweighted and multi-edge transformation (MET) methods.

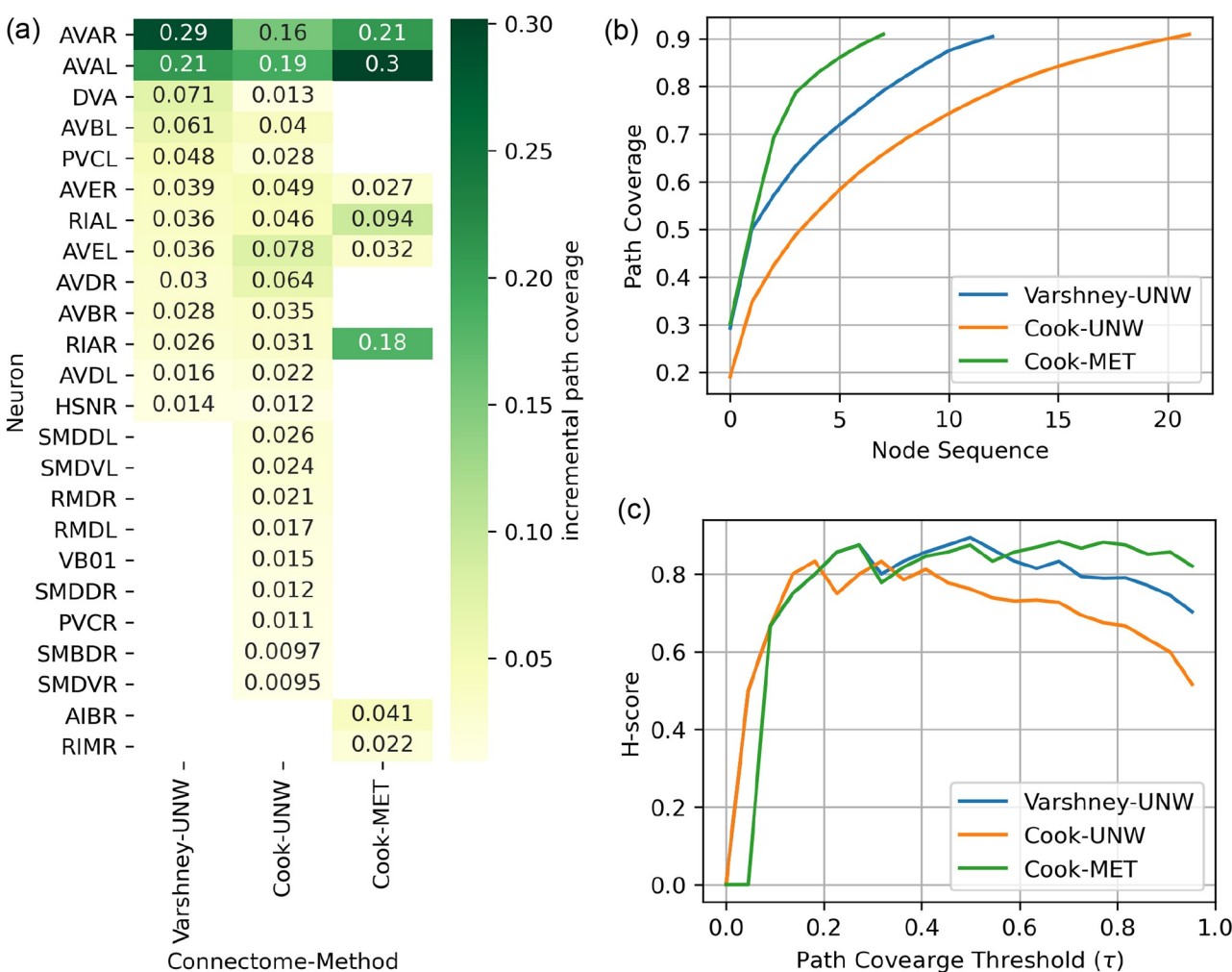

**Fig 3. Comparison of hourglass effects in *hermVarshney, hermCook* with or without MET transformation.** UNW: Unweighted version of hourglass analysis. MET: Mutli-edge transformation method of hourglass analysis for weighted networks. As described in subsections *Hourglass analysis on unweighted networks* and *Hourglass analysis on weighted networks*, hourglass analysis was performed to compute (a) the set of core nodes (neurons) in the $\tau$-core at $\tau = 0.9$ along with their path coverage as fraction of S-T paths that pass through the node, (b) Cumulative path coverage with the addition of each node in decreasing order of path coverage, and (c) the H-score metric at various values of $\tau$ to quantify the extent to which the network shows the hourglass effect.

generated by this routing scheme, the main aim of the analysis is to compute the smallest set of nodes (neurons) that cover a fraction $\tau$ of all the source-target (sensorimotor) paths in *P*. The path centrality (or coverage) metric is defined in *Hourglass analysis on unweighted networks* subsection to quantify the importance of a given node (or set of nodes) in the S-T information flow pathways in the network. To measure the extent to which a network exhibits the hourglass effect in its structure, the H-score is computed as detailed in subsection *Hourglass analysis on unweighted networks*. This subsection deals with comparison between the weighted (MET method) and unweighted (UNW) versions of the hourglass analysis for the *C. elegans* connectome, while the comparison between the hermaphrodite and male sexes is done in the subsequent subsection (*Sex-wise comparison of the hourglass effect*).

Fig 3 summarizes the results for the hourglass analysis using the unweighted (UNW) framework on *hermCook* and *hermVarshney* connectomes, as well as the weighted hourglass

framework (MET) on the *hermCook* connectome. As detailed below, the weighted version differs from the unweighted version in terms of various properties in the hourglass analysis.

In both weighted and unweighted cases, the networks do show the hourglass effect with a high H-score (Fig 3(c)). It can be noted in Fig 3(c) that with additional weaker edges, the H-score using UNW method for *hermCook* is much lower than that for *hermVarshney*, while the H-score for MET framework is much higher than that of the UNW framework on *hermCook*. Even though the UNW analysis on *hermCook* seems to indicate a much weaker hourglass effect, taking weights into account in the MET framework uncovers a much stronger hourglass structure in the same connectome (*hermCook*).

To identify the neurons that form the important nodes in the hourglass structure of the network, the $\tau$-core was computed as a set of iteratively selected nodes in the network that have the highest path coverage (see subsection *$\tau$-core and path centrality* and Algorithm 1). The nodes in the $\tau$-core can be considered to form the 'waist' of the hourglass structure in the network. Comparing the $\tau$-core for the UNW and MET cases (Fig 3(a)), the MET method results in a more specific set of core nodes than the UNW case for *hermCook*, while still retaining the set of essential high-centrality nodes. Moreover, the nodes corresponding to the pair of neurons RIAR and AVAL/R show much higher hourglass effect in the case of weighted analysis than in the case of unweighted analysis. The UNW method for *hermCook* identifies a much broader set of nodes compared the the *hermVarshney* connectome, which could be attributed to the additional edges in the *hermCook* connectome with weaker weights, thus weakening the hourglass structure. However, as is observable in the cumulative path coverage plots as well (Fig 3(b)), the weighted version of the analysis (MET) identifies a more specific set of nodes with much higher path coverage. In the case of the weighted hourglass analysis using MET method, the curve has steeper slope than in other cases, implying that certain nodes have much higher path centrality in the weighted analysis and less nodes are needed to cover the same fraction $\tau$ of all the weighted S-T paths.

Thus, when taking the weights into account, a stronger hourglass structure in the network can be unravelled, consisting of a smaller yet not fully overlapping set of core nodes compared to unweighted analysis. The MET method has the highest H-score for larger values of $\tau$ ($0.5 < \tau < 1$) despite the lowest values for UNW method on the same *hermCook* dataset. This further indicates that a framework considering the connectome as a weighted network reveals very different information about the hourglass properties of the network structure.

## Sex-wise comparison of the hourglass effect

To study the differences between the hourglass properties of the hermaphrodite (*hermCook*) and male *maleCook* connectomes, the results from weighted hourglass analysis using the MET method were analyzed in terms of their core neurons (i.e., nodes in the $\tau$-core at $\tau = 0.9$), cumulative path coverage and the H-score metrics. The connectomes *maleCook* and *hermCook* are visualized in Fig 4 with node-size representing the path centrality of the nodes as computed using the multi-edge transformation (MET) method. Fig 5 summarizes these results for comparison between the hermaphrodite and male connectomes. It can be noticed in Fig 5(a) that the set of nodes in the $\tau$-core have four common nodes between the two connectomes, corresponding to the neurons RIAL/R and AVAL/R which are ring and ventral chord inter-neurons located in the head, respectively. However, the hermaphrodite connectome has four unique nodes in its $\tau$-core corresponding to the neurons AVEL/R, AIBR and RIMR all of which are inter-neurons located in the lateral ganglia in the head, while the male connectome has five unique nodes corresponding to the neurons PVZ, PVV, PDB, PVX and PHCR. These neurons are mainly located in the pre-anal ganglion and involved in various functions related to mating

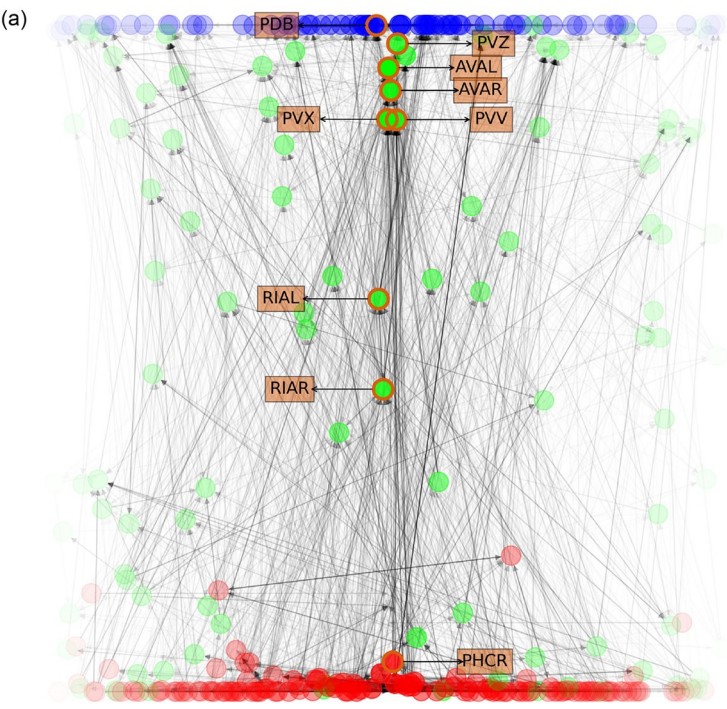

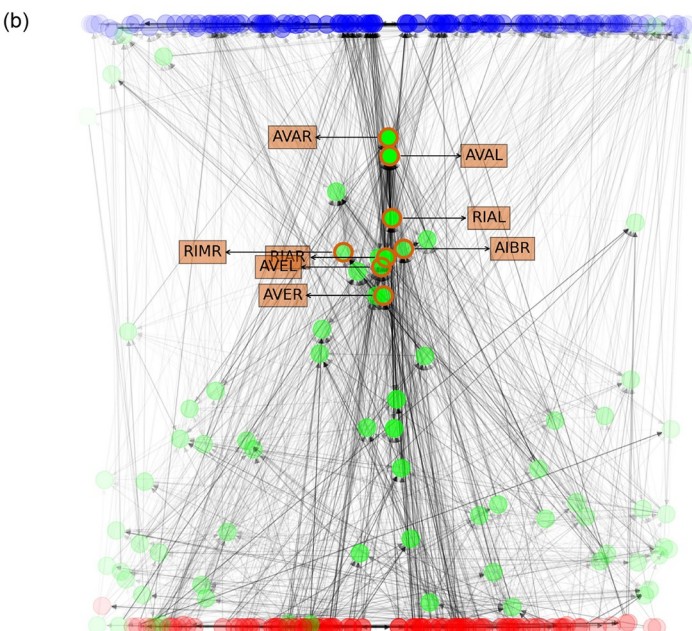

**Fig 4. A visualization of (a) *hermCook* and (b) *maleCook* connectomes.** Each node represents a neuron in the connectome, with its color representing the type of neuron (red: sensory neuron, green: inter-neuron, blue: motor neuron), and its transparency representing the path centrality calculated using the MET method. The vertical position of nodes are calculated based on the location metric in [25], which measures a node's relative position in the feedforward network. Horizontally, nodes are placed in the manner that nodes with higher path centrality are closer to the center. Each edge represents the collection of chemical synapses from one neuron to another, and the transparency indicates the count of chemical synapses in collection. The nodes forming the $\tau$-core (at $\tau = 0.9$) as described in subsection *Hourglass analysis on weighted networks* are labeled with their names.

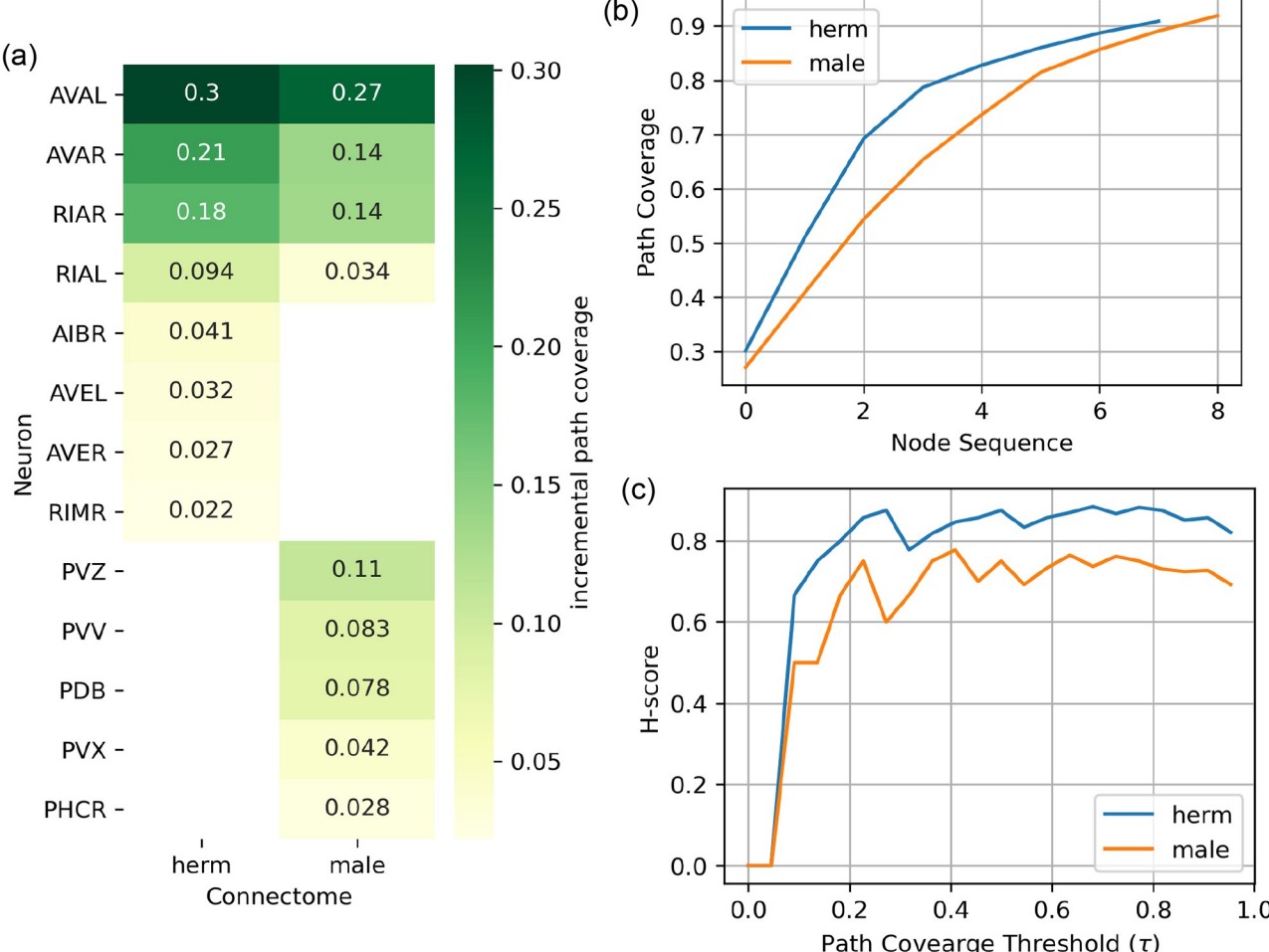

**Fig 5. Sex-wise comparison of the hourglass effects on *maleCook, hermCook* connectomes.** The multi-edge transformation (MET) was used as described in subsection *Hourglass analysis on weighted networks* to compute (a) the set of core nodes (neurons) in the $\tau$-core at $\tau = 0.9$ and their path coverage as fraction of S-T paths passing through the node, (b) Cumulative path coverage with the addition of each node in decreasing order of path coverage, and (c) the H-score metric to quantify the extent to which the network shows the hourglass effect.

including posture control, vulva detection and copulation [31]. These results are detailed in Table 4.

## Comparison with random networks

To check whether or not the hourglass effect is due to random effects, a comparison with randomized networks was done. The weighted connectomes were randomized by shuffling the edge weights, while keeping the set of edges the same to ensure that the structure of the network is maintained. Weighted hourglass analysis using the MET method was done on a set of 500 such networks and H-scores were computed for each of them at the same path-coverage threshold as used for the original networks ($\tau = 0.9$). Such a comparison between the H-score of the original connectomes with the corresponding set of randomized connectomes for each of the sexes can tell whether the hourglass effect observed is because of random effects simply due to the network structure or because of a more specific pattern of weights and S-T paths that leads the network to exhibit the hourglass property. It should be noted that for unweighted

**Table 4. Neurons in the τ-core at τ = 0.9 in the weighted hourglass analysis on *hermCook* and *maleCook* connectomes.**

| Neuron | τ-core membership | Category | Location | Function(s)* |
|---|---|---|---|---|
| RIAL/R | herm, male | inter-neuron | Head | Integration of outside information and inner state, behavioral response. |
| AVAL/R | herm, male | inter-neuron | Lateral ganglia of head | Command inter-neuron, locomotion |
| RIML/R | herm | inter-neuron | Lateral ganglia of head | Integration of outside information and inner state, behavioral response; locomotion |
| AVEL/R | herm | inter-neuron | Lateral ganglia of head | Command inter-neuron, backward locomotion |
| AIBR | herm | inter-neuron | Lateral ganglia of head | Integration of information from amphid sensory neurons; locomotion; information processing |
| PDB | male | inter-neuron | Pre-anal ganglion | Control mating posture |
| PVX | male | inter-neuron | Pre-anal ganglion | Backing during vulva search in mating behavior |
| PVZ | male | motor neuron | Pre-anal ganglion | copulation, vulva detection, spicule retraction |
| PVV | male | motor neuron | Pre-anal ganglion | Control mating posture |
| PHCR | male | motor neuron | Lumbar Ganglia | Temperature avoidance response |

The τ-core as defined in subsection *τ-core and path centrality* is the smallest set of nodes (neurons) in the network through which at least a fraction τ of source-target paths pass. It can be noticed that the four neurons common to the τ-cores of both the sexes are located in the head and are mainly invovled in the integration of information and locomotion. However, the neurons unique to the hermaphrodite are located in the head serving functions like information integration, behavioral response, locomotion etc, while most of the neurons unique to the male are located in the pre-anal ganglion and are involved in various functions related to mating.

* Functions of neurons were inferred from https://www.wormatlas.org/

networks, the randomization procedure carried out by [25] suffices as a check, but it is not the minimal randomization in the case of weighted networks.

The difference between H-score for the 500 random networks and the real network is statistically significant (with p-value $<10^{-2}$) for both the male and hermaphrodite connectomes when using the MET method as well as the UNW method. Fig 6 shows the distribution of the H-score curves of randomized networks and original network.

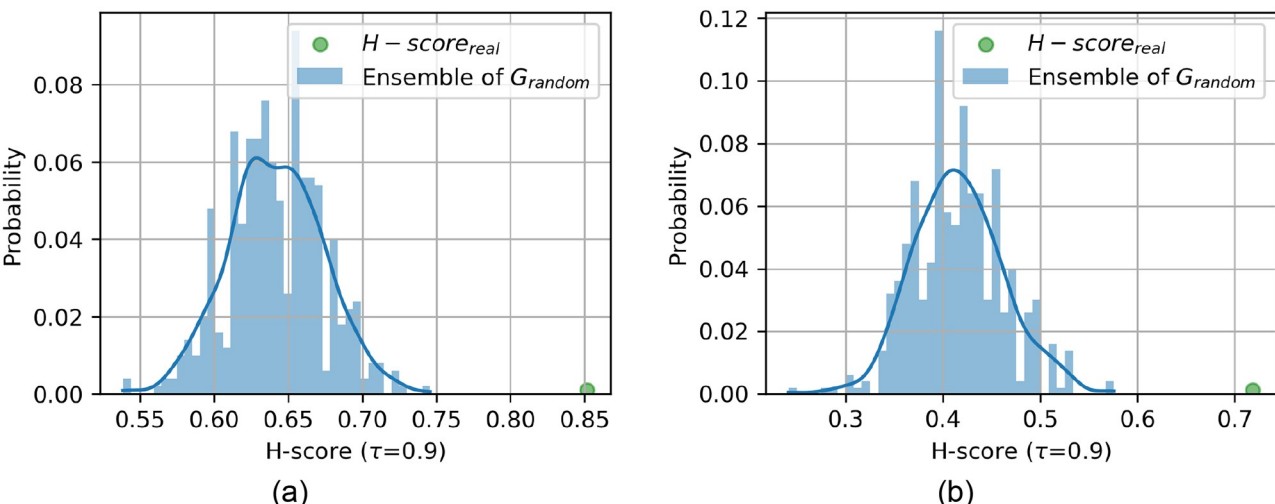

(a)    (b)

**Fig 6. Hourglass scores of the empirical network against networks generated with edge weight permutation.** For testing the results of the MET method in both (a) *hermCook* and (b) *maleCook* connectomes, only the edge weights were shuffled keeping the edge structure intact to generate 500 randomized networks in each case. Hourglass analysis on in all the above cases reveals significantly lower (*p*-value $<10^{-2}$) H-score than the corresponding original network.

## Discussion

In this work, we perform a comparison between the male and hermaphrodite structural connectomes of the *C. elegans* by developing a framework for studying the hourglass effect in weighted networks. Until recently, the *C. elegans* connectome was available only as an unweighted network for hermaphrodite. But with the recent advances [26], the connectome has been mapped for both the sexes along with edge weights based on the number and size of the synapses as seen with electron microscopy (EM). Our results show that there is a significant difference between the hourglass properties in the male and hermaphrodite connectomes, both in terms of the set of nodes that form the core of the network as well as the extent to which the network shows the hourglass effect.

Moreover, the hourglass analysis framework for weighted networks is able to narrow down the core nodes to a much smaller number of nodes in the network, which could mainly be because of taking the edge-weights into consideration in the computation of the centrality of the source-target pathways in the network. We had used five different networks created from two datasets. The widely used dataset by [23] only has the unweighted connectome for the hermaphrodite, while the remaining four networks were the weighted/unweighted networks from the male/hermaphrodite connectomes published by [26]. Among these connectomes, the variation within the same sex based on the dataset and choice of analysis (MET vs UNW) can be observed in Fig 3. In terms of the core nodes, the multi-edge transformation (MET) method for hourglass analysis results in a much smaller set of core nodes compared to the unweighted version (UNW) for both the sexes (Fig 5(a)). Additionally, the cumulative path coverage and H-score plots (Fig 3(b) and 3(c)) also indicate a similar trend that in the weighted paradigm of the analysis, the hourglass effect is more visible in the weighted connectomes as compared to the unweighted case. This could primarily be because of the enhancement of the centrality of certain neuronal pathways due to their predominant use in the organism as compared to others, thus leading to a higher number and size of synaptic connections between the participating neurons. This trend of significantly higher connectivity (and synaptic connections) of certain neurons and their pathways has been observed previously [32]. It can be further observed that the hourglass metrics (cumulative path coverage and H-score) are higher for the unweighted analysis (UNW) on the older dataset by [23] compared to the same analysis on the unweighted connectomes by [26] (See Fig 3(b) and 3(c)). The additional set of edges provided in the connectomes by [26] are significantly weaker compared to the older connectome by [23] (Fig 1), which could explain the emergence of a weaker hourglass effect when using the unweighted analysis (UNW). However, the same metrics under the weighted analysis (MET) are higher for the dataset by [26] indicating that the weighted framework is able to take into account the existence of numerous weaker connections which could otherwise change the result in case of the unweighted version of the analysis.

The comparison between the male and hermaphrodite connectomes reveals interesting patterns about how the neuronal networks in both the sexes are organized. While the hermaphrodite shows a stronger hourglass effect than the male, they have the same core size containing 9 neurons each (Fig 5(a)). However, in the case of the male connectome, the sex-specific neurons form the majority of the core nodes. This is in line with the observations from previous studies that the pruning of synapses, that occurs at sexual maturation, affects a lot of sensorimotor pathways leading to a sexually dimorphic neuronal connectivity in the organism [33]. Moreover, previous work has found that the posterior nervous system in the males gets enlarged to support mating behavior, adding about 30% more neurons with parallel pathways and complex connectivity, matching that of the whole nervous system of the hermaphrodite [34]. Our results additionally show that apart from just the addition of the sex-specific subnetwork, the

sexual dimorphism in the *C. elegans* also results in significantly different set of neurons forming the core of the sensorimotor information processing pathways in the connectome.

Through this work, we highlight a method to analyze a given weighted network for its hourglass property using an algorithmic approach to quantify its extent. The application of the framework to the weighted *C. elegans* connectomes for both the sexes reveals differences in the composition of the neurons that form the core of the hourglass structure in both the networks. Additionally, we show that when taking edge weights into account using the multi-edge transformation (MET) hourglass framework, different properties of the network emerge in terms of the core nodes as well the scale of the hourglass effect.

Various methods have been used previously for identifying critical neurons in the *C. elegans* connectome, such as graph perturbation techniques [35] and controllability frameworks [36]. The proposed framework in our paper is applicable only to directed networks with a given set of paths from source nodes to target nodes. The network may include cycles and the nodes may be of the same or different type—the hourglass method is applicable as long as there is a given set of directed paths from source to target nodes. A more extensive comparison between the hourglass framework and other related methods, such as core-periphery or rich-club approaches [37–40] can be found in [19].

More broadly, there are many approaches to identify important nodes (or rank them) in general networks [18, 39, 41]. The hourglass analysis framework is fundamentally different from approaches that are degree-based (such as k-core decomposition), that only consider shortest paths (such as betweenness centrality), or that consider all possible paths (potentially weighted based on their length, such as Katz centrality). A comparative study of existing and extended frameworks along with an interpretation of the outcomes should be done in the near future to enhance the understanding of the nature of optimization followed by naturally occurring neuronal networks.

## Acknowledgments

The authors are thankful to Prof. Scott Emmons from Albert Einstein College of Medicine, Bronx, NY, USA for help with clarifications on the *C. elegans* connectome dataset.

## Author Contributions

**Conceptualization:** Ishaan Batta, Constantine Dovrolis.

**Data curation:** Ishaan Batta, Qihang Yao.

**Formal analysis:** Ishaan Batta, Qihang Yao.

**Funding acquisition:** Constantine Dovrolis.

**Investigation:** Ishaan Batta, Qihang Yao, Constantine Dovrolis.

**Methodology:** Ishaan Batta, Qihang Yao, Kaeser M. Sabrin, Constantine Dovrolis.

**Project administration:** Constantine Dovrolis.

**Resources:** Constantine Dovrolis.

**Software:** Kaeser M. Sabrin.

**Supervision:** Constantine Dovrolis.

**Validation:** Ishaan Batta, Qihang Yao, Constantine Dovrolis.

**Visualization:** Qihang Yao.

**Writing – original draft:** Ishaan Batta.

**Writing – review & editing:** Ishaan Batta, Qihang Yao, Constantine Dovrolis.

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
