## [Decision Letter · Decision Letter 0]

7 Jun 2021

PONE-D-21-11096

A Weighted Network Analysis Framework for the Hourglass Effect - and its Application in the C. Elegans Connectome

PLOS ONE

Dear Dr. Batta,

Thank you for submitting your manuscript to PLOS ONE. After careful consideration, we feel that it has merit but does not fully meet PLOS ONE’s publication criteria as it currently stands. Therefore, we invite you to submit a revised version of the manuscript that addresses the points raised during the review process.

Based on the comments received from the reviews and my own observation I recommend minor revisions for the paper.

We look forward to receiving your revised manuscript.

Kind regards,

Thippa Reddy Gadekallu

Academic Editor

PLOS ONE

Journal Requirements:

We note that you have stated that you will provide repository information for your data at acceptance. Should your manuscript be accepted for publication, we will hold it until you provide the relevant accession numbers or DOIs necessary to access your data. If you wish to make changes to your Data Availability statement, please describe these changes in your cover letter and we will update your Data Availability statement to reflect the information you provide.

Reviewers' comments:

Reviewer's Responses to Questions

**Comments to the Author**

1. Is the manuscript technically sound, and do the data support the conclusions?

Reviewer #1: Yes

Reviewer #2: Yes

2. Has the statistical analysis been performed appropriately and rigorously? 

Reviewer #1: Yes

Reviewer #2: Yes

3. Have the authors made all data underlying the findings in their manuscript fully available?

Reviewer #1: Yes

Reviewer #2: Yes

4. Is the manuscript presented in an intelligible fashion and written in standard English?

Reviewer #1: Yes

Reviewer #2: Yes

5. Review Comments to the Author

Reviewer #1: This paper is a bio-informatics paper related to neural science. It develops a new framework for hourglass analysis that takes networks weights into account while

identifying the core nodes and the extent of hourglass effect in a given weighted network. The framework is used to study the structural connectome of the C.elegans and identify intermediate neurons that form the core of sensori-motor

pathways in the organism.

There are three types of neurons in hourglass analysis on the C. elegans connectome, including sensory neuron, inter-neuron and moter neuron. The paper develops a multi-edge transformation (MET) to deal with the unweighted/weighted networks. For model details, it selects core nods with large path centrality scores and defines Hourglass score as a criteria to retrieve the best hourglass property of a network. It runs a lot of experiments with random setups and sex-wise comparison to see the hourglass effect under different scenarios.

The overall paper is well written and with essential research values. The only advice is that as the network contains multiple node neuron types and is directed. it would be better if the authors can mention some interpretations or comparisons for homogeneous/heterogeneous and (un)directed networks. This content can be addressed throughout analysis or in the future work.

Reviewer #2: The authors have done a good job and addressed all the comments and suggestions. The paper seems to be in a good shape now. I recommend that the paper can be accepted for publication in the current form.

6. PLOS authors have the option to publish the peer review history of their article (what does this mean?). If published, this will include your full peer review and any attached files.

Reviewer #1: **Yes: **Zheng Gao

Reviewer #2: No

---

## [Author Response · Author response to Decision Letter 0]

28 Aug 2021

We are very thankful to the editor and the reviewers for taking out the time to review and provide valuable feedback on our work. We have worked to incorporate the suggestions made in the letter for minor revision. Please find in the Response Letter a point-by-point response to the suggestions from both editor and the reviewers. Any references to page numbers, sections, figures, and papers refer to the new revised and marked manuscript file.

---

## [Decision Letter · Decision Letter 1]

9 Sep 2021

A Weighted Network Analysis Framework for the Hourglass Effect - and its Application in the C. Elegans Connectome

PONE-D-21-11096R1

Dear Dr. Batta,

We’re pleased to inform you that your manuscript has been judged scientifically suitable for publication and will be formally accepted for publication once it meets all outstanding technical requirements.

Kind regards,

Thippa Reddy Gadekallu

Academic Editor

PLOS ONE

Additional Editor Comments (optional):

Reviewers' comments:

Reviewer's Responses to Questions

**Comments to the Author**

1. If the authors have adequately addressed your comments raised in a previous round of review and you feel that this manuscript is now acceptable for publication, you may indicate that here to bypass the “Comments to the Author” section, enter your conflict of interest statement in the “Confidential to Editor” section, and submit your "Accept" recommendation.

Reviewer #1: All comments have been addressed

2. Is the manuscript technically sound, and do the data support the conclusions?

Reviewer #1: Yes

3. Has the statistical analysis been performed appropriately and rigorously? 

Reviewer #1: Yes

4. Have the authors made all data underlying the findings in their manuscript fully available?

Reviewer #1: Yes

5. Is the manuscript presented in an intelligible fashion and written in standard English?

Reviewer #1: Yes

6. Review Comments to the Author

Reviewer #1: I think the authors have addressed all my previous concerns. This version should be in the ready status to get accepted.

7. PLOS authors have the option to publish the peer review history of their article (what does this mean?). If published, this will include your full peer review and any attached files.

Reviewer #1: No

---

## [Editor Report · Acceptance letter]

18 Oct 2021

PONE-D-21-11096R1 

A Weighted Network Analysis Framework for the Hourglass Effect — and its Application in the C. Elegans Connectome 

Dear Dr. Batta:

I'm pleased to inform you that your manuscript has been deemed suitable for publication in PLOS ONE. Congratulations! Your manuscript is now with our production department. 

Kind regards, 

on behalf of

Dr. Thippa Reddy Gadekallu 

Academic Editor

PLOS ONE